# Joining the Conversation: Towards Language Acquisition for Ad Hoc Team Play

**Dylan R. Cope**
King's College London
dylan.cope@kcl.ac.uk

**Peter McBurney**
King's College London
peter.mcburney@kcl.ac.uk

## Abstract

In this paper, we propose and consider the problem of *cooperative language acquisition* as a particular form of the *ad hoc team play* problem. We then present a probabilistic model for inferring a speaker's intentions and a listener's semantics from observing communications between a team of language-users. This model builds on the assumptions that speakers are engaged in *positive signalling* and listeners are exhibiting *positive listening*, which is to say the messages convey hidden information from the listener, that then causes them to change their behaviour. Further, it accounts for potential sub-optimality in the speaker's ability to convey the right information (according to the given task). Finally, we discuss further work for testing and developing this framework.

## 1 Introduction

Typically, in the field of *emergent communication* a group of agents learn to interact with one another through communication channels in order to facilitate coordination in a shared environment, i.e. a Dec-POMDP (Oliehoek & Amato, 2016). The agents learn highly effective communication strategies, but they tend to be brittle in the sense that they are unable to coordinate with agents that they have not encountered before. This construction does not naturally lend itself to systems that require a machine to communicate with a human, or enter within a community of humans using language to coordinate. In this paper, we frame this as a problem of *cooperative language acquisition*, where the goal is to adopt the language of a community of agents so as to coordinate with them.

More precisely, we place the problem in the context of *ad hoc team play* (Stone et al., 2010). In ad hoc team play, we are given a set of *competent*[1] agents and a domain of coordination tasks, and the problem is to design new agents that are capable of achieving success when playing with randomly sampled teammates. In our problem, we assume that there exists a community of language-users that define the pool of players who, by means of their shared language, are all successful ad hoc team players. Therefore, the cooperative language acquisition problem is defined as the case of designing a new agent to join this pool of players by observing a sample of interactions from the community.

## 2 Background

A *decentralised partially-observable Markov decision process* (Dec-POMDP) is described by a 7-tuple $(\mathcal{S}, \{\mathcal{A}_i\}, T, R, \{\Omega_i\}, O, \gamma)$, where $\mathcal{S}$ is a set of states, $\{\mathcal{A}_i\}$ is a set of action sets, $T$ is a transition function, $R$ is a reward function, $\{\Omega_i\}$ is a set of observation sets, $O$ is an observation function, and $\gamma$ is a discount factor. We will talk of *trajectories* for a given agent $i$, which are sequences of state-action-reward tuples $\tau \in \mathcal{T}_i = (\mathcal{S} \times \mathcal{A}_i \times \mathbb{R})^*$. Each agent $i$ follows a policy $\pi_i$ that maps an observation sequence to a distribution over actions. We denote the distribution over future trajectories that a policy induces as $\pi(\tau|.)$. The *return* of a trajectory is computed as the discounted sum of rewards: $V(\tau) = \sum_{k=0}^{|\tau|} \gamma^k r_k$

---

[1]Meaning that they can achieve some threshold of success in all environments, given a fixed team.

Following Lowe et al. (2019), we suppose that each agent's action sets can be expressed as $\mathcal{A}_i = \mathcal{A}_i^c \cup \mathcal{A}_i^e$, where $\mathcal{A}_i^c$ is a set of *communicative actions* and $\mathcal{A}_i^e$ is a set of *environment actions*. Communicative actions are sent to a target agent by a dedicated cheap-talk channel (there is no cost to communication), meaning they appear in the receiver's observation at the next time step. We also use Lowe et al.'s (2019) definitions of *positive listening* and *positive signalling*:

**Definition 1** (Positive Listening). *An agent $i$ with the policy $\pi_i$ exhibits positive listening if there exists a message generated by a signaller $j$, $m \in \mathcal{A}_j^c$, such that $d_\tau(\pi_i(\tau|z, \mathbf{0}), \pi_i(\tau|z, m)) > 0$ where $\mathbf{0}$ is a zero vector, $z$ is a variable that conditions the policy (e.g. observations and/or latent memory), and $d_\tau$ is a distance metric over $\mathcal{T}_i$.*

**Definition 2** (Positive Signalling). *Let $m = (m_0, \ldots, m_T)$ be a sequence of messages sent by an agent over the course of a trajectory of length $T$, and similarly for observations $o = (o_0, \ldots, o_T)$, and actions $a = (a_0, \ldots, a_T)$. An agent exhibits positive signalling if $m$ is statistically dependent on either $a$ or $o$.*

## 3 One-Way Communication Problem Formulation

We start to formulate the problem of cooperative language acquisition with the simplest case involving two agents: a *speaker $A$* and a *listener $B$*. For a particular interaction $i$ the speaker emits a message $m_i \in \mathcal{A}_A^c{}^*$ that is received by the listener, and then the listener takes actions that lead it on a trajectory $\tau_i \in \mathcal{T}_B$ in a Dec-POMDP sampled from a given domain. We are only considering the case of *one-way communication* with this set-up, but we will discuss two-way communication, i.e. dialogues, in Section 6. To make this more precise, we assume that $A$ and $B$ are agents sampled from a pool of players operating in an ad hoc team, where the domain $D$ is a set of *referential games*, i.e. a class of Dec-POMDPs based on Lewis signalling games (Lewis, 1969; Lazaridou et al., 2017; Lee et al., 2018). We will assume that the listener is exhibiting *positive listening* to the messages sent by the speaker, and the speaker is *positive signalling* an 'intended' *target trajectory*[2], $\tau_i^\odot \in \mathcal{T}_B$. We can denote this by saying that the observer observes: $m_i \sim \pi_A(m \mid \tau_i^\odot)$ and $\tau_i \sim \pi_B(\tau \mid m_i)$, where $\pi_A$ and $\pi_B$ denote the policies followed by each agent respectively.[3]

Before moving on, let us define a running example game to illustrate the setting. Suppose that the speaker has access to a shopping list and a map of the supermarket, and must write a note for the listener to observe, who then must retrieve the items as quickly as possible.

The cooperative language acquisition task is to construct an agent $X$, who we will call the *observer*, which is able to take on the roles of either the speaker or the listener and successfully communicate with others. So, if $X$ is taking on the role of the speaker, given some $\tau$ that they intend for $B$ to follow, they should emit a message that maximises the probability that $B$ does so. If $X$ is acting as the listener, and receives some $m \sim \pi_A(m \mid \tau^\odot)$ they should estimate $\tau^\odot$ and follow this trajectory. With this, given a dataset of interactions between speakers and listeners, $m_i, \tau_i \sim \mathcal{D}_{AB}$, we can define the following sub-problems:

**Problem 1** (The Forward Problem (Signalling)). *Find a function $\beta(m \mid \tau, \theta_\beta)$ parameterised by $\theta_\beta$, which we call the Broca function, such that $m$ maximises the probability that the listening agent $B$ follows the trajectory $\tau$ upon receiving $m$, i.e.:*

$$\theta_\beta^* = \arg\max_{\theta_\beta} \sum_{\tau \in \mathcal{T}_B} E_{m \sim \beta(m|\tau, \theta_\beta)}\big[\pi_B(\tau \mid m)\big] \tag{1}$$

**Problem 2** (The Backward Problem (Listening)). *Find a function $\nu(\tau \mid m, \theta_\nu)$ parameterised by $\theta_\nu$, which we call the Wernicke function. Given the message $m$ is from the speaking agent $A$ and is intended to invoke the trajectory $\tau^\odot$, the function $\nu$ should maximise the probability of $\tau^\odot$.*

$$\theta_\nu^* = \arg\max_{\theta_\nu} \sum_{\tau^\odot \in \mathcal{T}_B} E_{m \sim \pi_A(m|\tau^\odot)}\big[\nu(\tau^\odot \mid m, \theta_\nu)\big] \tag{2}$$

---

[2]This definition of positive signalling is slightly different to that of Lowe et al. (2019) as we are referring to trajectories rather than actions/observations.

[3]We are also assuming that all the participants are sincere in their communications and actions, with none trying to deceive others.

## 4 Finding Broca and Wernicke

Firstly, we can directly model the forward problem with the data available. We estimate the parameters $\theta_\beta$ by mapping from observed trajectories to messages received by $B$:

$$\theta_\beta^* = \arg\min_{\theta_\beta} \sum_{m_i, \tau_i \in \mathcal{D}_{AB}} d_m(m_i, \hat{m}_i) \tag{3}$$

$$\text{where } \hat{m}_i = \arg\max_m \beta(m \mid \tau_i, \theta_\beta)$$

Given a distance function $d_m$ over messages. Put differently, we are aiming to find $\theta_\beta$ such that the Broca function can produce the message that caused a given trajectory in the data. To place this into our running example, we have data regarding the notes that were sent to the shopper $(m_i)$, and paths through the shop that the shopper took $(\tau_i)$, and we are learning the relationship between notes and paths.

However, the backward problem is much trickier given that we are never able to directly observe the intended trajectory $\tau_i^\odot$ for the message $m_i$ sent by the speaker. If we assume that the speaker is optimal, i.e. it always sends the perfect message to invoke the intended actions in $B$, then $\tau_i^\odot = \tau_i$ and thus we can optimise the reverse mapping as in the forward problem (i.e. messages to trajectories). But what can we do if we wish to relax this?

Instead of modelling the speaker as perfectly optimal, we can assume 'soft-optimality', otherwise known as *Boltzmann-rationality*[4]. We will do this in two parts: first, we will assume that given the target trajectory $\tau^\odot$, the speaker is more likely to send messages that are 'closer to optimal', for which we need some notion of *semantic distance* between messages. Secondly, we will assume that the speaker is more likely to pick target trajectories for the listener that yield a high return in the Dec-POMDP. Put formally:

$$P(\tau^\odot) \propto \exp(V(\tau^\odot)) \tag{4}$$

$$P(m|\tau^\odot) \propto \exp(-\mathbb{S}_B(m_B^*(\tau^\odot),\ m)) \tag{5}$$

Where, $V$ is the expected return of a given trajectory, $m_B^*(\tau^\odot)$ is the optimal message to send to $B$ to maximise the chance that $B$ takes the trajectory $\tau^\odot$, and $\mathbb{S}_B$ is a measure of the semantic distance between two messages for $B$. These latter two are defined as follows:

$$m_B^*(\tau) = \arg\max_m \pi_B(\tau|m) \tag{6}$$

$$\mathbb{S}_B(m_1, m_2) = d_\tau\big(\pi_B(\tau \mid m_1), \pi_B(\tau \mid m_2)\big) \tag{7}$$

In other words, the semantic distance is a function of the difference in actions that $B$ takes (characterised by the distance function over trajectories $d_\tau$) as a result of different messages. Thus, it is mathematically similar to Lowe et al.'s (2019) definition of positive listening, and philosophically close to the various approaches to 'meaning' that couple information and action (Haig, 2017; Wittgenstein, 1953; Peirce, 1878). Additionally, note that if we substitute these definitions into $P(m|\tau^\odot)$:

$$P(m|\tau^\odot) \propto \exp\Big(-d_\tau\big(\pi_B(\tau \mid \arg\max_m \pi_B(\tau^\odot|m)), \pi_B(\tau \mid m)\big)\Big) \tag{8}$$

$$\propto \exp\Big(-d_\tau\big(\tau^\odot, \pi_B(\tau \mid m)\big)\Big)$$

Thus the expression involving the semantic distance captures the intuition that the more optimal messages are the ones that, in expectation, lead to trajectories that are closer to the target. With our assumptions in place, we now insert this into the backward problem. For a given interaction $m_i, \tau_i \sim \mathcal{D}_{AB}$ we can express most probable target trajectory $\hat{\tau}_i^\odot$ with the *maximum a posteriori* estimate:

$$\hat{\tau}_i^\odot = \arg\max_{\tau^\odot} P(\tau^\odot \mid m_i) = \arg\max_{\tau^\odot} P(\tau^\odot)P(m_i \mid \tau^\odot)$$

$$= \arg\max_{\tau^\odot} \big(V(\tau^\odot) - \alpha d_\tau(\tau^\odot,\ \pi_B(\tau \mid m_i))\big) \tag{9}$$

$$= \arg\max_{\tau^\odot} \big(V(\tau^\odot) - \alpha d_\tau(\tau^\odot,\ \tau_i)\big)$$

---

[4]See Jeon et al. (2020) for an overview of Boltzmann-rationality and its advantages for modelling human decision-making.

Where $\alpha$ is a hyperparameter controlling our prior on the relative optimality of the speakers ability to effectively communicate versus pick the optimal trajectory (similar to Jeon et al. (2020)). Therefore, to estimate the parameters of the Wernicke function $\theta_\nu$:

$$\theta_\nu^* = \arg\min_{\theta_\nu} \sum_{m_i, \tau_i \in \mathcal{D}_{AB}} (\alpha d_m(\hat{\tau}_i^\odot, \tau_i) - V(\hat{\tau}_i^\odot))$$

$$\text{where } \hat{\tau}_i^\odot = \arg\max_\tau \nu(\tau \mid m_i, \theta_\nu) \tag{10}$$

Again, let us contextualise this within the example of the shoppers. If the speaker's note contained roughly the right set of instructions, but is perhaps slightly confusing in a way that throws off the shopper (perhaps impossible directions), the Wernicke function will not emulate the shoppers confusion. Instead, the estimate of the intended trajectory can take into account the ambiguity or inconsistency and try and figure out what would be a successful path through the supermarket. Comparatively, when we assume optimality of the speaker, we are forced to conclude that the shoppers confusion was intended.

## 5 Related Work

A close area of related work is *inverse reinforcement learning* (Russell, 1998; Ng & Russell, 2000). Namely, the modelling of the speaker is similar to IRL, where instead of there being a hidden reward function influencing the agents' actions, there is a target trajectory for the listener. Further, the Boltzmann-rational model used is very similar to approaches in IRL (Zietbart et al., 2008; Finn et al., 2016). In the field of emergent communication, the works of Lee et al. (2018) and Lazaridou et al. (2017) both present frameworks for grounding learning agents in human natural language. They do this by using text annotated images rather than data from direct human communication in a cooperative setting. Finally, outside of AI, in the economics literature there has been work modelling how an uniformed listener may extract information from informed debaters (Glazer & Rubinstein, 2001). But because of markedly different assumptions, it does not tackle the problem of the current paper.

## 6 Discussion and Conclusion

In this paper we have presented the first steps towards constructing an agent that, given data regarding the interactions of language-users, can find the meaning behind messages received, as well as optimally convey a recommended trajectory to a listener. Yet, there are still several directions of further work to be explored.

Firstly, how do we extend the system to *dialogues*, i.e. two-way communication? Potentially the system naturally captures dialogues as both agents can play the roles of speakers and listeners simultaneously (or interchangeably). For instance, suppose that in the supermarket example the speaker and the shopper held a phone call, and the shopper asks a question for clarification on directions. The shopper does not have an exact intended trajectory for the speaker's response, because if they knew this they would not need to ask the question. However, this does not necessarily pose a problem for the framework presented in this paper. Although we have referred to the target trajectory as "intentional" it is not necessary for a speaker to know the full details of the trajectory. This applies so long as they help the listener to find the closest trajectory that maximises reward, which they may do so by adding their own private information.

Secondly, but no less critically, is the problem of empirically testing this framework by constructing an agent. There are several suitable test environments, for example, gridworld games that are similar to the supermarket example (Kajić et al., 2020; Leibo et al., 2017), or more communication focused problems such as the game of *Taboo*. In this game, one person has to get another to say a hidden word, but they are forbidden from revealing certain pieces of helpful information[5]. Finally, this work could be extended from the passive case of observing data, to a situation where the learner is engaged with the language-users, perhaps for example with an *active learning* approach (Settles, 2009).

---

[5]Taboo has already been proposed as an interesting challenge for AI (Rovatsos et al., 2017).

ACKNOWLEDGEMENTS

Work done by DC is thanks to the UKRI Centre for Doctoral Training in Safe and Trusted AI (EPSRC Project EP/S023356/1).

We would like to thank Francis Rhys Ward, Nandi Schoots, Richard Willis, Mattia Villani and Charles Higgins for their help.

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
