# OpenReview forum: "Joining the Conversation: Towards Language Acquisition for Ad Hoc Team Play"
_ICLR.cc/2022/Workshop/EmeCom — EmeCom Workshop at ICLR 2022_

### Official Review · Reviewer_NRDd · 2022-03-21

**Rating:** Strong Accept
**Confidence:** 4

**Review:**

The authors propose a simple (but non-classic from the EC community) model of communication. The paper is easy to follow with simple nice intermediate examples. The models also highlight some limitation of classic optimization of the lewis game (i.e. the lack of feedback).
I would have been curious to add even a toyish experiment to test the optimization procedure, and obtain an empirical insight of the system dynamics.
The paper perfectly match the EC workshop, and I am looking forward to a discussion with the authors. An excellent workshop paper!

---

### Official Review · Reviewer_Dy1f · 2022-03-23
**Probabistic framework to infer the latents of a community and learn a new compatible agent.**

**Rating:** Weak rejection
**Confidence:** 4

**Review:**

In this paper about cooperative language acquisition, the authors try to tackle the problem of trying to learn a new agent ( sender or listener ) which can adapt to a given community of sender or listener using a data of their interactions. They propose using a probabistic framework to infer the latents of a community and then learn a new compatible agent.

Pros
* The use of boltzman rationality to infer the latent $\tau^{\bigodot}$ from given $m, \tau$ is interesting. This can help in giving a inverse rl style insight into what the sender agent's intended objective was and help in accomplishing the task to maximize reward.

Cons
* A potential limitation is that it's possible that the whole community of agents have co-adapted with each other and the new agent that gets learned using the data from their interactions also overfits and hence doesn't generalize to actual human interactions. Ideas from  language drift and community regularization community might be helpful here.
* The authors restrict themselves to single-turn referencial games like lewis games which can be a big limitation although some idea are presented to address these concerns.
* It's not clear how this probabilistic approach would compare to the standard methods of using gumbel-softmax or RL fine-tuning in these referential games.
* The writing of the paper can be greatly improved. How the approach helps with the objective of the paper is not super apparent.

Comments
*  The authors mention that the agents can take the roles of speakers and listeners interchangeable but it's not clear to me how since the input/output spaces for both can be different.
* If I understand clearly this forward-backward framework can also be used to finetune a set of pre-trained sender-listener agents using  alternate optimization.

---

### Decision · Program_Chairs · 2022-03-25

**Decision:**

Accept

**Comment:**

Reviewers agree that the authors propose an interesting new framework for learning ad-hoc communication. There are some recommendations of improved baselines but both reviewers seem excited by the idea and we look forward to having the authors discuss this at the workshop and get important feedback on their work.